# The DAMA25 Study: Feasibility of a Lifestyle Intervention Programme for Cancer Risk Reduction in Young Italian Women with Breast Cancer Family History

**DOI:** 10.3390/ijerph182312287

**Published:** 2021-11-23

**Authors:** Giovanna Masala, Domenico Palli, Ilaria Ermini, Daniela Occhini, Luigi Facchini, Lisa Sequi, Maria Castaldo, Saverio Caini, Benedetta Bendinelli, Calogero Saieva, Melania Assedi, Ines Zanna

**Affiliations:** 1Clinical Epidemiology Unit, Institute for Cancer Research, Prevention and Clinical Network (ISPRO), 50139 Florence, Italy; b.bendinelli@ispro.toscana.it; 2Cancer Risk Factors and Life-Style Epidemiology Unit, Institute for Cancer Research, Prevention and Clinical Network (ISPRO), 50139 Florence, Italy; d.palli@ispro.toscana.it (D.P.); i.ermini@ispro.toscana.it (I.E.); danielaocchini@yahoo.it (D.O.); l.facchini@ispro.toscana.it (L.F.); l.sequiesterno@ispro.toscana.it (L.S.); maria_castaldo@hotmail.it (M.C.); s.caini@ispro.toscana.it (S.C.); c.saieva@ispro.toscana.it (C.S.); m.assedi@ispro.toscana.it (M.A.); i.zanna@ispro.toscana.it (I.Z.)

**Keywords:** familial breast cancer risk, young women, single-arm intervention, diet, physical activity

## Abstract

Background: Diet and physical activity (PA) can modulate sporadic and possibly familial breast cancer (BC) risk. The DAMA25 study is a single-arm 12-month intervention aimed to modify dietary and PA habits in healthy young Italian women with a positive BC family history, categorized as having intermediate or high genetic risk according to NICE (National Institute for Health and Cancer Excellence) guidelines. Methods: Participants, aged 25–49 years, were asked to adopt a diet mainly based on plant-based foods and to increase moderate daily activities combined with 1 h/week of more intense activity. Cooking lessons, collective walks, educational sessions, brochures, booklets and online materials were implemented. Dietary, PA habits and anthropometry were collected at baseline and at the end of the intervention. Changes on dietary, lifestyle habits and anthropometry were evaluated by GLM adjusted for weight reduction counselling aimed to participant with a BMI ≥ 25, age and baseline values of each variable. Results: Out of 237 eligible women 107 (45.2%) agreed to participate and among them 98 (91.6%) completed the intervention. The adherence rate of the intervention was 77.8%. We observed a reduction in red and processed meat (*p* < 0.0001) and cakes consumption (*p* < 0.0001). Consumption of whole grain bread (*p* < 0.001), leafy vegetables (*p* = 0.01) and olive oil (*p* = 0.04) increased. We observed an increase in moderate (*p* < 0.0001) and more intense (*p* < 0.0001) recreational activities, an average 1.4 kg weight loss (*p* = 0.005), a reduction of waist circumference (*p* < 0.001) and fat mass (*p* = 0.015). Conclusions: The DAMA25 study shows that it is feasible an intervention to improve in the short-term dietary and PA habits and anthropometry in women with high BC familial risk.

## 1. Introduction

Breast cancer (BC) is the most common female cancer worldwide. In Italy about 50.000 women were estimated to be diagnosed with BC in 2019 and 25% of these tumours occurred before age 50 [1]. Overall, 20–25% of all BC cases are associated with a family history (FH) of breast and other cancers, and up to 10% may be linked to germinal mutations in susceptibility genes, mainly BRCA1 or BRCA2 [2,3].

Lifestyle and environmental factors have been supposed to modulate BRCA1/2 mutation penetrance and also familial BC risk [4,5,6,7]. It is well known that body mass index (BMI) is positively associated with BC risk in postmenopausal women, while it tends to be inversely associated with risk in pre-menopausal women [8,9].

Recently, it has been shown that the higher a woman’s familial risk is, the greater is the influence of BMI on her absolute postmenopausal BC risk [9]. Moreover, in women with a BRCA mutation the maintenance of a healthy weight and a weight loss between the ages 18 and 30 were associated with a reduction in BC risk [6]. Therefore, maintaining a healthy weight is important for women with a BC positive FH. Interestingly, in a large familial BC cohort, the adherence to American Cancer Society recommendations on BMI, PA and alcohol intake was associated with a lower overall mortality [10].

Clinical services offered to women with a BC/OC FH include risk assessment and regular surveillance for early diagnosis of BC and recurrences. Genetic testing is proposed when FH suggests the presence of a mutation in BRCA genes or other major genes implicated in familial BC. Prophylactic surgery may be proposed as a risk reduction strategy but its acceptance is not widespread and varies in different populations [11]. On the other hand, women enrolled in these surveillance programs are often eager to take individual action to reduce their risk, and the risk of other family members, notably their daughters. 

Most FH clinics offer only general advice, but the possibility to contribute to lifestyle modifications during counselling sessions and diagnostic follow-up, should be considered [12]. 

There is the need to plan and evaluate specific behavioural intervention programs tailored on high-risk subjects including those with a familial BC risk, taking into account that the earlier lifestyle changes are introduced the more effective is the intervention.

We carried out the DAMA25 study with the aim of evaluating the feasibility of a structured programme of dietary and PA modifications in young Italian healthy women with a positive BC FH. The combined PA and dietary intervention were designed to modify lifestyle habits of participants without a specific focus on weight reduction, although in overweight/obese women some specific advice was offered. 

## 2. Materials and Methods

The DAMA 25 study (ISRCTN54262307) is a single-centre project aimed to evaluate the feasibility of a single-arm 12-month behavioural intervention to modify diet and PA level in young women with a familial BC risk.

The study was approved by the Ethics Committee of the local health authority in Florence (Italy).

### 2.1. Selection of Study Participants and Data Collection

Participants were healthy women aged 25–49 years residing in the Florence area, with a positive BC FH assessed in the frame of the Genetic Counselling Service of the Institute for cancer research, prevention and clinical network (ISPRO), in Florence.

We invited to participate pre-menopausal women without a previous diagnosis of cancer, diabetes or other major co-morbidities such as major cardiovascular or neurological diseases able to hamper their active participation in the study.

Eligible women were invited to attend a small group meeting where the programme was presented. Women who agreed to participate were given an appointment for the baseline visit and were asked to fill 2 validated questionnaires previously used in the frame of the EPIC-Italy study to collect information on their baseline dietary and PA habits [13].

The food frequency questionnaire (FFQ), comprised 14 sections concerning 188 different food items. The times a given food item was consumed (per day, week, month, or year) was requested. The quantity of the food consumed was assessed by selection of an image of a food portion, or applying a standard portion. Dietary data were then converted into average daily quantities of foods [14]. The lifestyle questionnaire (LSQ) included information on education level, reproductive and medical history and a section on PA. PA at work was classified as follows: sedentary, standing, manual or heavy manual work. Information on hours spent in leisure time includes recreational activities such as walking (a moderate activity including walking to the work place, shopping, walking for pleasure), biking, and more intense fitness activities (including gym activities, swimming, playing tennis, running, etc.) and household activities (do-it-yourself activities, gardening and house cleaning) [15].

### 2.2. Baseline Visit 

At the baseline visit, women signed the study informed consent form and returned the questionnaires. Weight, height, hip and waist circumference were measured by trained personnel. Body composition was measured with a Tanita Multi-frequency Segmental Body Composition Analyser MC-780 MA (Tanita, Arlington Heights, IL, USA). A fasting venous blood sample was taken from each woman within 1 month from the visit in the period between the 3rd to the 11th day of the menstrual cycle and stored in the biological bank of the project. 

### 2.3. WCRF Score Construction

At baseline, an index score reflecting concordance with the first seven 2018 WCRF/AICR (World Cancer Research Fund/American Institute for Cancer Research) recommendations for cancer prevention was constructed [16,17]. The assigned score was 1, 0.5 and 0 when the recommendation was met, partially met and not met. A single score for each participant was calculated adding the scores obtained for each component. The maximum score was 7 points.

The score was further categorized into 3 categories according to predefined cut-off: category 1 (<3 points), category 2 (≥3 to <5 points), category 3 (≥5 points).

### 2.4. Dietary Intervention 

During an initial individual counselling session, study dieticians assessed each woman’s dietary habits by the questionnaire filled at baseline, explained the dietary intervention, and provided suggestions on how to put the requested changes into practice. Each woman was asked during an individual counselling to adopt a diet mainly based on plant-based foods, with a low glycemic load, low in saturated and *trans-fats* and alcohol, and rich in antioxidants. 

The following suggestions based on WCRF/AICR guidelines were given:cereals (bread, pasta, and grains): gradual replacement of refined grains with whole grains, in particular whole-wheat bread. Increase in whole grain cereals (such as rice, spelt and barley).not starchy vegetables: consumption of one portion of raw and one portion of cooked vegetables at each meal including sauces for pasta based on tomatoes, zucchini, artichokes, broccoli, etc.; vegetable and legume soups, addition of vegetables to meat dishes and sandwiches.fish: consumption at least biweekly. Both fresh and frozen, not precooked fish were acceptable choices.meat and processed meat: consumption of fresh and processed red meat had to be reduced to less than 1 time per week considering all types. Poultry (chicken/turkey) was suggested as an acceptable alternative (with a maximum of 2 times per week).legumes and pulses: their consumption had to be gradually increased to at least three-four portions/week.fruit: consumption of 2 to 3 portions of fruit per day including fruit at breakfast or as snacks during the day, as an alternative to cookies and pastries.added fat: high-quality extra virgin olive oil had to be used as the only dressing and cooking fat. Its use for cooking dishes had, however, to be reduced, while no restrictions were made for seasoning vegetables, legumes, soups, etc.processed foods: to be reduced to a minimum, decreasing the consumption of ready-to-eat dishes, desserts, cookies, processed meat, etc.sugar, sweets, desserts: consumption had to be reduced or avoided. Advice was given to replace sweets with fruit.milk and dairy products: limited consumption, exclusion of full-fat varieties.wine and spirits: consumption of no more than 1 glass of wine per day, at meals, if already used to drink wine. The consumption of other alcoholic beverages and of ready-to-eat dishes was discouraged.

Participants were requested to attend two meetings and six cooking classes during the study period, both to be held in groups of approximately 25 women. During cooking classes conducted by a professional cook and a dietician, dishes were prepared and consumed jointly at the end of each session.

Group meetings were organised at 1st and 3rd month of the intervention focusing on: diet for disease prevention, nutritional value of foods, energy balance and readiness to change. 

Cooking classes, organized at 2nd, 4th, 6th and 8th month of the intervention, were led by a professional cook and a study dietician in an appropriate facility. During these classes, dishes mainly based on whole cereals, vegetables and legumes were prepared in agreement with the traditional local cooking and the recommendations that the women had received, showing the ingredients and seasonings that had to be preferably used in the preparation of meals. Prepared dishes were consumed jointly at the end of the session. Examples of recipes and daily and weekly menus were also provided to participants.

### 2.5. Physical Activity Intervention

Each woman discussed changes in PA levels requested by the study protocol and how to adapt them to her daily schedule during an initial individual counselling. 

We aimed to increase moderate daily recreational activities, such as walking at moderate pace and biking corresponding to 3 MET-hours/day (MET = metabolic equivalent) up to 1 h/day, to be combined with at least 1 h/week of more intense activity accounting for 6–10 MET-hours/week [18]. Suggested activities to be included in the daily routine were walking at moderate pace, biking and slow dancing. Women were also given a specific equipment set to exercise at home (a soft ball, an elastic band, two dumbbells, a gym mat), and at 6th month of the intervention 10 video-tutorials (lasting 7–10 min), specifically developed by our team, suggesting structured fitness more intense activities. Study participants were invited at 1st and 3rd month of the intervention to participate to 2 educational meetings about the benefits of PA, and to 2 group walks at 2nd and 4th month of the intervention supported by the study PA team and were also motivated to organize autonomously periodic group walks.

In the ISPRO website, a specific DAMA25 section was periodically updated with dedicated materials (weekly menu, selected recipes, itineraries for walks).

Overweight/obese women (BMI ≥ 25 kg/m^2^) were offered, in addition, specific advice on the quantity of food to be consumed and individualized advice to increase PA gradually. 

Study participants were requested to keep 8 written 1-week diaries on diet and PA (a diary approximately every 40 days). The study personnel reviewed the diaries so that they could monitor by a phone call the achievement of the study objectives and discuss with each participant what still remained to be done and to agree on a plan to overcome any difficulties.

### 2.6. Final Visit

All participants were invited to a visit 12 (±3) months after their enrolment in the study. During the final visit, the same protocol of the baseline visit was applied including anthropometric measures, FFQ and LSQ questionnaires and blood samples. Then participants filled in a questionnaire using a score between 1 and 5 to rate their satisfaction level and the perceived efficacy of the material and activities proposed. 

### 2.7. Statistical Analysis

Changes on dietary, lifestyle habits and anthropometry were evaluated by GLM adjusted for weight reduction counselling aimed to participant with a BMI ≥ 25 (yes/no), age (continuous) and baseline values of each variable (continuous). A *p* value < 0.05 was considered statistically significant. All analyses were performed using SAS software (version 9.2) (SAS Institute Inc., Cary, NC 27513, USA). Results are presented for the 98 women who completed the protocol.

## 3. Results

Subjects were recruited between January and March 2016, with 107 participants (45%) among 237 eligible women. (Figure 1). Information collected at baseline and at the end of study were available for 98/107 (91.6%) participants and the mean interval between the two visits was 13.5 months (SD 15.9).

### 3.1. Family History

Study participants had been categorized as having intermediate (53.3%, 57/107) or high genetic risk (46.7%, 50/107) according to NICE international guidelines [19].

The most frequently reported tumours in family members were BC and OC, while the most frequent affected relative was the mother. Three participants reported a male BC FH. 

Seventy-eight percent (39/50) of participants classified as high-risk had at least one BC/OC affected first degree relative (mean age at diagnosis 45.1 years, 41.0% diagnosed at an age < 40 years,) nine (23.1%) reported a diagnosis of bilateral BC and three (7.0%) a relative with both BC and OC. Overall, 8 high-risk study participants carried a mutation in BRCA1 or BRCA2 susceptibility gene (16%).

Ninety-four percent (54/57) of intermediate-risk women reported at least one BC affected first degree relative (mean age at diagnosis 52.0 years; 14.8% diagnosed at an age ≤40 years), and five (9.3%) a relative with a diagnosis of bilateral BC. No relative with a diagnosis of OC was reported in this group. 

No BC was diagnosed among participants during the study period.

### 3.2. Baseline Characteristics

The mean age of participants was 41.7 year (SD 5.7). The mean value of BMI was 24.5 (SD 5.6) and 31.6% of participants were overweight/obese. Most of them were married (65.3%), had at least one child (68.4%) and, among these, 91.0% had breast fed. Fifty-one percent reported to have a university degree and among women with a paid work (92.8%) most reported a sedentary work (72.5%). Most participants were never smokers (56.1%) while for current smokers (22.5%) the mean number of cigarettes/day was 8. About eighty-six percent had used or were currently using contraceptive pill (Table 1).

### 3.3. WCRF Adherence

The distribution at baseline of DAMA25 study participants within the 3 defined categories of the WCRF single score is reported in Table 2. The mean score was 3.64 (SD 1.05: min 1.0 max 5.75). Individuals with a lifestyle according to WCRF recommendations (score ≥ 5) were 13.09%, most of participants had an intermediate score (≥3 and <5; 63.55%) and 23.36% had the lowest score (<3). Table 3 shows the distribution at baseline of DAMA25 study participants by single recommendation. Notably, 48.6% of participants were physically inactive and 58.88% had an excessive consumption of red and processed meat.

### 3.4. Compliance with the Proposed Interventions

At the end of the intervention, participants reported a borderline statistically significant increase in the consumption of total vegetables and soups (*p* = 0.06) and a 20% increase in the consumption of leafy vegetables (*p* = 0.01).

The consumption of bread and other bakery products slightly decreased, while whole-grain bread consumption increased (*p* < 0.001). We observed a significant decrease in the reported consumption of red and processed meat (−51.5%; *p* < 0.0001), poultry (−36.0%; *p* < 0.0001), cakes (−37.3% *p* < 0.0001), milk (−26.7%; *p* < 0.001) and cheese (−38.2%; *p* <0.0001). Overall, we observed an increase in legumes (65.4%) and fish (44.9%) consumption although these results were not significant (Table 4). An increase in the consumption of olive oil (8.9%; *p* = 0.04), nuts, seeds and dried fruit (*p* < 0.02) and a decrease in wine consumption (−13.0%; *p* < 0.0002) also emerged. 

A significant increase of 3.6 h/week for overall leisure time PA emerged (*p* = 0.0002). In particular, we registered an increase in all recreational activities (*p* < 0.0001), including walking (*p* < 0.0001) and fitness (*p* < 0.0001) (Table 5). 

Participants experienced an average weight loss of 1.4 Kg (*p* = 0.005) and an average reduction of waist circumference of 3.6 cm (*p* < 0.001). Weight reduction (−3.40 Kg) was concentrated in overweight/obese women (*n* = 31) while waist decreased in all women. We also observed a significant reduction of fat mass (*p* = 0.015) (Table 6), more evident in women at counselling (−2.67 kg). 

### 3.5. Adherence to Protocol and Satisfaction Questionnaire

Overall, the adherence to the proposed protocol activities was 77.8%. In particular, participation to practical activities (cooking classes and collective walks) was 64.4% and to educational meetings was 75.6%. The percentage of compilation of the 1-week diaries was 86.2% for dietary and 84.6% for PA diaries.

About 88% (86/98) of participants declared to be very satisfied of the study experience. Cooking classes were judged very useful (score 4 or 5) by 83.3%, collective walks by 66.2%, educational meetings by 85.7%, PA diaries by 75.5% and dietary diaries by 79.5% of participants. Eighty-eight percent of participants reported to have used the equipment set to exercise at home and 51% the video-tutorials. About 88% of participants visited the DAMA25 section of the ISPRO website. The vast majority (90%) of participants reported that familial members were interested in the program. 

## 4. Discussion

The results of the DAMA25 study support the feasibility of an intervention aimed to change dietary and PA habits in healthy young women with a family history of BC, combining both practical activities and educational sessions. 

At baseline most of our participants showed an acceptable level of adherence to the WCRF recommendations for cancer prevention. However, considering the scores obtained for each component the need for a lifestyle intervention that primarily promoted an increase in PA levels and a reduction in the consumption of red and processed meat emerged. 

The majority of women who accepted to participate completed the study and reported changes in their dietary and PA habits at least in a short-term period.

We observed a reduction in the consumption of red and processed meat. Consumption of whole-grain bread also increased, replacing white bread, while consumption of sweet foods was reduced. We recorded an increase in the consumption of leafy vegetables and olive oil. The consumption of red and processed meat is considered a probable risk factor for several tumours including breast cancer also among premenopausal women [7,20,21,22]. At the same time an increased intake of wholegrains, vegetables, fruit, and legumes seems to protect against cancer including breast cancer [7,23].

We observed a significant increase in the time dedicated to PA both in relation to moderate activities and to more intense recreational and sporting activities, the most relevant for risk reduction in young women [24]. Overall, we observed a slight reduction in weight and fat mass and an improvement in central obesity indices, which are considered indicators of increased risk for several cancers including BC [25]. These anthropometry modifications were obtained although the intervention was not specifically aimed to a weight reduction, except for obese participants.

Recently, other studies have been implemented to evaluate the possibility to change lifestyle habits in high-risk young women [26,27]. Among them the LivingWell study demonstrated that a lifestyle program for people with a FH of BC cancer is feasible and the results suggest favorable outcomes in terms of PA increase and reduction in dietary fat [26]. The importance of offering to women with a BC positive FH also a counseling on potentially lifestyle modifiable factors has been debated [28,29]. Familial and hereditary tumours, although rare in the general population, have a strong psycho-emotional impact both for the affected patients and for their healthy family members, and have significant implications at the clinical-diagnostic (surveillance, early diagnosis) and preventive level (chemo-prevention interventions, prophylactic surgery). Indeed, women with familial BC are motivated to take action to modify their risk. Notably the proportion of women who accepted to participate to DAMA25 study was quite high despite the effort required. The intervention program, including informative and practical activities, was positively valuated by the participants. In young women, busy with work and/or with children, one of the limitations in changing lifestyle could be the difficulty in integrating them into family life. In this sense the online materials could have been a valid help, in fact the DAMA25 home page was visited regularly by half of the study participants who considered it a very useful tool, in addition to brochures and booklets.

FH clinics usually offer only general advice about lifestyle. This intervention model or some of its tools might be used in clinical centres dealing with family risk assessment or regular surveillance where indications for lifestyle changes could be conveyed with more efficacy. 

Moreover, this intervention model could be applied in research projects aimed to evaluate, with an appropriate sample size and design, the effect of lifestyle changes on cancer risk or intermediate biomarkers. Intervention studies evaluating the impact of lifestyle factors on familial BC risk have been recently implemented [30,31]. 

During the intervention we collected blood samples at baseline and after one year in order to be able in the future, to evaluate specific biomarkers and their possible changes after one year of treatment. 

Finally, some of the tools developed in the DAMA25 study could also be made available to different groups of the general population as young adults.

A limitation of our study is that DAMA25 is a single-arm study so we are unable to make a comparison with a control group. In addition, most of the DAMA25 participants had a high educational level and this could have positively affected their motivation to participate and to follow the recommendations. Finally, given the limited number of subjects, it was not possible to carry on subgroup analyses.

## 5. Conclusions

Our findings suggest that women with BC family history, if properly stimulated, are capable of improving their lifestyle and that our procedures and tools could be useful to promote a healthy lifestyle as a primary preventive intervention and a risk reducing strategy in these high-risk women. The possibility to maintain these changes over time and the way to introduce these aspects in a larger clinical setting need to be further evaluated.

## Figures and Tables

**Figure 1 ijerph-18-12287-f001:**
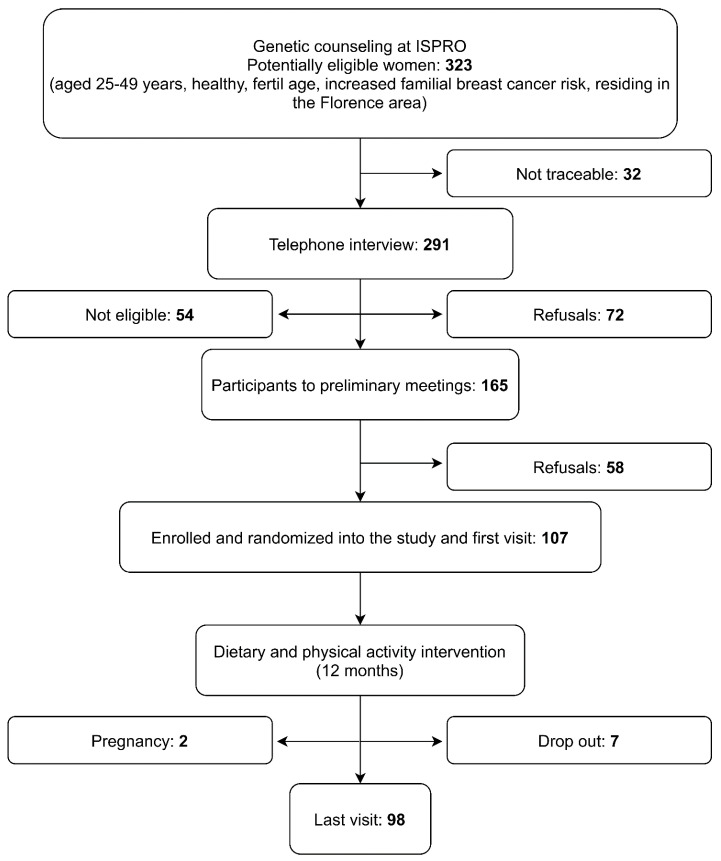
Flow chart of the DAMA25 Study (Trial Registration Number: ISRCTN54262307).

**Table 1 ijerph-18-12287-t001:** Distribution according to selected variables of the 98 women enrolled into the DAMA25 study (The DAMA25 study, Florence, Italy, 2016–2017).

	*n* (%)
**Age (years)**	
<40	31 (31.6)
≥40	67 (68.4)
**BMI (kg/m^2^)**	
<25	67 (68.4)
≥25	31 (31.6)
**Living with a partner**	
No ^1^	34 (34.7)
yes	64 (65.3)
**Number of children**	
0	31 (31.6)
1	30 (30.6)
≥2	37 (37.8)
**Breastfeeding ^2^**	
no	6 (9.0)
yes	61 (91.0)
**Education**	
Primary school	3 (3.1)
High school	45 (45.9)
University	50 (51.0)
**Paid work**	
no	7 (7.2)
part-time	31 (31.6)
full time	60 (61.2)
**Physical activity at work**	
sedentary	66 (72.5)
standing	15 (16.5)
manual	10 (11.0)
**Smoking history**	
Current	22 (22.5)
Former	21 (21.4)
Never	55 (56.1)
**Contraceptive pill use**	
Current	14 (14.3)
Ever	71 (72.4)
Never	13 (13.3)
**Total**	**98**

^1^ Including widows and divorced women. ^2^ 67 women reporting at least 1 child.

**Table 2 ijerph-18-12287-t002:** Distribution at baseline of the 107 DAMA25 study participants according to the 2018 WCRF/AICR adherence score in categories.

2018 WCRF/AIRC Adherence Score	Absolute Frequency (N)	Relative Frequency (%)
^1^ Category 1	25	23.36
^1^ Category 2	68	63.55
^1^ Category 3	14	13.09

^1^ WCRF/AICR adherence score: category 1: <3; category 2: (≥3 and <5); Category 3: (≥5).

**Table 3 ijerph-18-12287-t003:** 2018 WCRF/AICR recommendations for cancer prevention and operationalization of the WCRF/AICR score in the 107 participants of DAMA25 study.

2018 WCRF/AIRC Recommendations	Operationalisation of Recommendations	Points	Absolute Frequency (N)	Relative Frequency (%)
**1. Be a healthy weight**	**BMI (kg/m^2^):**			
	<18.5 or ≥30	0	15	14.02
	25–29.9	0.25	19	17.76
	18.5–24.9	0.5	73	68.22
	**Waist circumference (cm(in)):**			
	≥88 (≥35)	0	20	18.69
	80–<8 (31.5–<35)	0.25	16	14.95
	<80 (31.5)	0.5	71	66.36
**2. Be physically active**	**Total moderate-vigorous physical** **activity (min/wk):**			
	<75	0	52	48.60
	75–<150	0.5	17	15.89
	≥150	1	38	35.51
**3. Eat a diet rich in** **wholegrains, ** **vegetables, fruit and** **beans**	**Fruits and vegetables (g/day):**			
	<200	0	9	8.41
	200–<400	0.25	36	33.64
	≥400	0.5	62	57.94
	**Total fiber (g/day):**			
	<15	0	19	17.76
	15–<30	0.25	77	71.96
	≥30	0.5	11	10.28
**4. Limit consumption** **of “fast food” and other** **processed food high in** **fat, starches or sugars**	**Percent of total Kcal from ultra-** **processed foods (aUPFs):**			
	Tertile 1	0	36	33.64
	Tertile 2	0.5	36	33.64
	Tertile 3	1	35	32.71
**5. Limit consumption** **of red and processed** **meat**	**Total red meat (g/wk) and processed** **meat (g/wk):**			
	Red meat >500 or processed meat ≥100	0	63	58.88
	Red meat <500 and processed meat 21–<100	0.5	34	31.78
	Red meat <500 and processed meat <21	1	10	9.35
**6. Limit consumption** **of sugar-sweetened ** **drinks**	**Total sugar-sweetened drinks** **(g/day):**			
	>250	0	1	0.93
	>0–≤250	0.5	81	75.70
	>250	1	25	23.36
**7. Limit alcohol** **consumption**	**Total ethanol (g/day):**			
	>14 (1 drink)	0	12	11.21
	≤14 (1 drink)	0.5	85	79.44
	0	1	10	9.35

**Table 4 ijerph-18-12287-t004:** Estimated changes in consumption of selected food (g/day) and kilocalories at the end of the intervention in comparison to baseline. (The DAMA25 study, Florence, Italy, 2016–2017).

Food Groups (g/day)	Baseline Mean Median (10–90°)	Follow up Mean Median (10–90°)	Difference ^1^ (%)	*p* Value ^2^
Total vegetables and soups	230.0 215.7 (115.2–373.6)	256.8 225.3 (141.7–413.3)	26.8 (11.7)	0.06
- leafy vegetables	40.4 35.4 (10.6–79.3)	48.0 38.7 (17.3–85.2)	7.6 (18.8)	0.01
Legumes	21.1 17.2 (4.3–44.9)	34.9 32.0 (9.2–66.4)	13.8 (65.4)	0.39
Fresh fruit	262.8 230.2 (104.9–474.9)	273.2 243.6 (136.3–434.6)	10.4 (4.0)	<0.0001
Nuts, seeds and dried fruit	2.8 1.4 (0.2–7.3)	4.8 3.7 (0.2–10.5)	2.0 (71.4)	0.02
Bread	114.2 101.7 (19.0–150.4)	97.2 80.3 (18.9–203.4)	−17.0 (14.9)	<0.0001
White bread	42.3 24.3 (0.0–100.00)	12.5 3.6 (0.0–44.4)	−29.8 (70.5)	<0.0001
Whole grain bread	34.5 15.9 (0.0–100.0)	65.1 43.6 (5.1–177.8)	30.6 (88.7)	<0.0001
Red and processed meat	61.7 57.2 (16.3–116.1)	29.9 27.0 (4.9–61.2)	−31.8 (51.5)	<0.0001
Poultry	33.3 29.4 (2.8–76.7)	21.3 16.3 (0.8–44.6)	−12.0 (36.0)	<0.0001
Fish	41.2 39.8 (7.9–71.7)	59.7 60.1 (19.7–102.6)	18.5 (44.9)	0.16
Cakes and cookies	97.0 89.5 (35.4–174.9)	60.8 51.8 (26.4–99.8)	−36.2 (37.3)	<0.0001
Olive oil	27.0 27.4 (13.8–43.2)	29.4 25.9 (16.8–45.8)	2.4 (8.9)	0.04
Seed oil	0.4 0.3 (0.0–0.4)	0.3 0.3 (0.0–0.4)	−0.1 (25.0)	<0.0001
Butter	1.3 0.4 (0.0–2.9)	0.6 0.4 (0.1–1.1)	0.7 (53.8)	<0.0001
Milk	87.6 34.3 (0.0–224.0)	64.2 14.6 (0.0–164.0)	−23.4 (26.7)	<0.0001
Cheese	48.9 43.2 (11.0–99.8)	30.2 24.8 (10.6–55.0)	−18.7 (38.2)	<0.0001
Wine ^3^	49.1 17.9 (2.1–125.0)	42.7 17.9 (4.2–125.0)	−6.4 (13.0)	<0.0002

^1^ Difference between the end of intervention and the baseline data. ^2^ *p*-values were calculated by GLM adjusted for counselling (yes/no), age in continuous and baseline continuous values of each variable. ^3^ Only for 83 drinkers.

**Table 5 ijerph-18-12287-t005:** Estimated changes in leisure time PA (hours/week) at the end of the intervention (The DAMA25 study, Florence, Italy, 2016–2017).

Leisure Time Physical Activity (Hours- Week)	Baseline Mean Median (10–90°)	Follow up Mean Median (10–90°)	Absolute Difference ^1^ (%)	*p* Value ^2^
Overall leisure time physical activity	16.52 10.8 (3.9–24.6)	20.13 14.4 (5.5–31.3)	3.61 (21.9)	0.0002
- Household physical activity	10.57 10.6 (3.5–24.8)	11.45 10.8 (3.9–24.6)	0.88 (8.3)	0.004
- Recreational physical activity	5.95 4.6 (1.3–10.8)	8.67 8.3 (3.3–15.0)	2.72 (45.7)	<0.0001
Walking	3.37 2.5 (0.5–7.5)	5.13 4.5 (1.5–9.5)	1.76 (52.2)	<0.0001
Fitness	1.75 0.8 (0.0–5.5)	2.55 1.5 (0.0–5.5)	0.80 (45.7)	<0.0001

^1^ Difference between the end of intervention and the baseline data. ^2^
*p*-values were calculated by GLM adjusted for counselling (yes/no), age in continuous and baseline continuous values of each variable.

**Table 6 ijerph-18-12287-t006:** Estimated changes in anthropometric measures at the end of the intervention in comparison to baseline (The DAMA25 study, Florence, Italy, 2016–2017).

	Baseline Mean Median (10–90°)	Follow up Mean Median (10–90°)	Absolute Difference ^1^ (%)	*p* Value ^2^
Weight (kg)	65.86 62.6 (51.0–81.6)	64.42 62.3 (50.3–80.1)	−1.44 (2.2)	0.005
Waist (cm)	79.20 76.0 (67.6–3.50)	75.60 73.0 (65.0–88.0)	−3.60 (4.5)	<0.001
Fat mass ^3^ (kg)	20.00 17.7 (10.7–31.6)	18.80 17.2 (10.3–30.7)	−1.17 (5.9)	0.015
Lean mass ^3^ (kg)	45.79 44.7 (39.9–53.5)	45.54 45.3 (39.2–52.6)	−0.19 (0.4)	0.24

^1^ Difference between the end of intervention and the baseline data. ^2^
*p*-values were calculated by GLM adjusted for age in continuous and baseline continuous values of each variable. ^3.^ For 1 woman the data was not available.

## Data Availability

The data presented in this study are available on request from the corresponding author. The data are not publicly available due to participants privacy protection.

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
