# Peer review of "The DAMA25 Study: Feasibility of a Lifestyle Intervention Programme for Cancer Risk Reduction in Young Italian Women with Breast Cancer Family History"

_ijerph, 2021, doi:10.3390/ijerph182312287_

Round 1
Reviewer 1 Report
I enjoyed reading this manuscript and learning about this intervention study in young women at high risk of breast cancer.
I have some general and specific comments and suggestions noted below.
General comments
This study incorporated a number of appropriate activities and resources that contributed to its success and perhaps long-lasting lifestyle changes for the participants. These included: having individual and group sessions; cooking classes; providing key resources like equipment for home use, video tutorials for exercises, recipes meeting dietary recommendations; organizing group walks; information on simple ways to adopt healthier diets and physical activity into daily routines; and online resources (suitable for this age group).
The interventions could be adapted to include family members or workplace interventions to reduce sedentary behaviour too.
This study yielded high quality data, body composition and biological sample collections that could be used to answer additional research questions.
Specific Comments by Section
Abstract: Define NICE before using acronym
Introduction
Was nicely written and structured, with relevant and recent publications cited. I suggest revising “in BRCA mutated women” to “women with a BRCA mutation.”
Methods
The target population, recruitment and details about interventions are generally well described. The study was a registered trial, which is an important strength. The use of validated questionnaires is another strength and could also enable comparison with the EPIC-Italy data at DAMA baseline collection.
It was not clear if the dietary recommendations by the dietician were based on results from the completed diet questionnaire at baseline or the first weekly food diary. In addition, the list of dietary suggestions did not seem to be based on a national food guide or cancer prevention guide although were appropriate for the dietary recommendations. There were no details on when the 8 weekly diaries were written or how they were used beyond study personnel reviewing them and monitoring progress.
Define WCRF/AICR before using acronym.
Results
Analyses were adjusted for baseline values and important confounders, such as weight counselling and age. The study had good uptake and retention rates for interventions in both physical activity and diet over a 1-year period. The results were clearly presented in multiple tables, although some presentation as figures (e.g., bar graphs) could be done as well.
The changes in anthropomorphic values stratified by BMI at baseline would be interesting to see, as it is likely the women with excess weight who are affecting the group mean/median changes over the course of the study.
Minor edits: Table 1b, typo in Point 5 about red meat; Table 2 – It is difference not absolute difference since some entries are negative. Last sentence is missing a ‘.’ after programme.
Discussion
I needed to look up the term heredo-familial as it was not familiar to me. One online dictionary (dictionary.com) stated is was no longer in technical use so perhaps consider using a more common term, such as inherited genetic disease as that might be clearer.
Some text could adopt active voice or be revised to improve clarity, for example:
“The intervention program, including informative and practical activities, was valuated in a positive way by the participants.” could be revised to “The intervention program, including informative and practical activities, was positively valued by the participants.”
Author Response
Please, see the attachment

Reviewer 2 Report
This is a single-arm, single-center intervention trial with a 12-month dietary and physical activity intervention in young women with family history of breast cancer. The study was well designed and conducted. This report is focused on the feasibility of the lifestyle intervention. The study achieved excellent retention (91.6%) and adherence (77.8%) and provided a preliminary efficacy of positive changes in lifestyle behaviors as well as body weight and fat mass. The findings of this study can be a foundation of future studies to investigate the effects of such lifestyle interventions in young women with higher risk of developing breast cancer.
1. Overall
- For the Title, please specify what “risk reduction” means (e.g., breast cancer risk?). Also, please consider indicating what this current manuscript is reporting (e.g., feasibility of …).
- Please consider replacing “plant foods” with “plant-based foods” or “plants” based on the context.
2. Abstract
- Was it a single-arm study? If so, please indicate it in the abstract and remove “non-randomized”.
- In Methods, please briefly mention how the intervention was delivered and how outcome assessments were measured.
- In Results, what is “more intense” activities? Did you mean the same “vigorous activity” mentioned in Methods?
- Please provide the adherence rate of the intervention.
- It would be difficult conclude the intervention had an effect on changing outcomes in this study as there was no comparison. The conclusion will need to focus on the feasibility of the intervention.
3. Methods
- Please also indicate that it was a single arm study.
- Regarding dietary intervention, at what time points did two meetings and six cooking classes occur within the 12-month intervention period?
- What did the specific equipment set for exercise include?
- Were the 10 video-tutorials watched at participants’ discretion or were they given to them one by one throughout the year? This is important because the frequency of the contacts from the staff can substantially impact the adherence and efficacy of the intervention.
- What is “collective” walk?
- Please also specify when (or at which time point of the intervention) the meetings and interactions with the study team occurred.
- The statistical analyses were not based on regression modelling to account for other potential significant covariates, which typically include BMI and/or smoking status. This should be discussed in the Discussion section as a limitation.
4. Results
- Can you provide a table for baseline demographic and socioeconomic characteristics of the study participants?
- In Table 1, please provide footnotes to briefly describe what each category means if some descriptions exist.
- It appeared in the abstract that you measured moderate and vigorous physical activity, but they are not shown in Table 3. If you measured physical activity by intensity, please provide the statistics in Table 3.
- In Table 3, what is “fitness” and how was it measured? This was not mentioned in the Methods section.
5. Discussion
- Please expand your discussion on the implications of the reductions in red and processed meat and the increase in plant-based diet, including why such foods should be avoided or encouraged.
- Please provide the limitations of the study.
Author Response
Please, see the attachment
